# Living Plants Ecosystem Sensing: A Quantum Bridge between Thermodynamics and Bioelectricity

**DOI:** 10.3390/biomimetics8010122

**Published:** 2023-03-14

**Authors:** Alessandro Chiolerio, Giuseppe Vitiello, Mohammad Mahdi Dehshibi, Andrew Adamatzky

**Affiliations:** 1Center for Converging Technologies, Bioinspired Soft Robotics, Istituto Italiano di Tecnologia, Via Morego 30, 16065 Genova, Italy; 2Unconventional Computing Laboratory, University of the West of England, Coldharbour Lane, Bristol BS16 1QY, UK; 3The Cyberforest Experiment, Costa Bocche, Località Paneveggio, 38037 Predazzo, Italy; 4Department of Physics “E.R. Caianiello”, Universitá degli Studi di Salerno, Via Giovanni Paolo II 132, 84084 Fisciano, Italy; 5Faculty of Computer Science, Multimedia and Telecommunications, Universitat Oberta de Catalunya, Rambla del Poblenou 156, 08018 Barcelona, Spain

**Keywords:** *Picea abies*, bioelectric potential, electrophysiology, quantum field theory, fractal dimension, Shannon entropy

## Abstract

The in situ measurement of the bioelectric potential in *xilematic* and *floematic* superior plants reveals valuable insights into the biological activity of these organisms, including their responses to lunar and solar cycles and collective behaviour. This paper reports on the “Cyberforest Experiment” conducted in the open-air Paneveggio forest in Valle di Fiemme, Trento, Italy, where spruce (i.e., *Picea abies*) is cultivated. Our analysis of the bioelectric potentials reveals a strong correlation between higher-order complexity measurements and thermodynamic entropy and suggests that bioelectrical signals can reflect the metabolic activity of plants. Additionally, temporal correlations of bioelectric signals from different trees may be precisely synchronized or may lag behind. These correlations are further explored through the lens of quantum field theory, suggesting that the forest can be viewed as a collective array of in-phase elements whose correlation is naturally tuned depending on the environmental conditions. These results provide compelling evidence for the potential of living plant ecosystems as environmental sensors.

## 1. Introduction

Plants have been studied for their salient intelligence [1,2,3,4,5], capabilities to implement distributed information processing [6,7,8,9], indications of advanced perception, cognition and adaptive behaviour [10,11,12,13], anticipatory responses [14], and swarm intelligence [15]. Higher plants employ impulses of electrical activity to coordinate the actions of their bodies and long-distance communication [16,17,18].

The bursts of impulses could be either endogenous [19], e.g., related to motor activities [20,21,22,23], or in a response to external stimulation, e.g., the temperature [24], osmotic environment [25], or mechanical stimulation [26,27]. Electrical signals could propagate between any type of cells in plant tissue, there are indications, however, of higher conductivity of the vascular system, which might act as a network of pathways for travelling electrical impulses [28,29,30,31].

In a previous study [32], we established a methodological framework for measuring, sorting, and analysing bioelectric data and demonstrated that forests could behave as collective entities using quantum field theory. In this study, we use higher-order statistics and quantum field theory to analyse the electrical activity of superior plants and their thermal fluctuations, gaining deeper insights into the complex interactions within forests and their impacts on the surrounding environment.

We present the results of the Cyberforest Experiment [32], which was conducted in the Paneveggio forest in Valle di Fiemme, under the town of Paneveggio (TN, Italy) [33,34,35,36], at an altitude of 1950 metres and covering an area of 8000 square metres. This research marks a significant advancement in our understanding of the complex and interconnected nature of the forest ecosystem and has the potential to revolutionise our knowledge of ecosystem functioning on a global scale. Consequently, the findings of this study will be of great interest to scientists and researchers across multiple disciplines, including ecology, environmental science, physics of complex systems, and quantum field theory.

The organisation of the paper is as follows: Section 2 presents the experimental setup and numerical analysis. Section 3 describes the electrical and thermographic measurements conducted on three individual trees/logs. Section 4 outlines the theoretical modelling of the forest as a collective organism using a quantum field theory, and finally our conclusions are drawn in Section 5. The Appendices contain details about the formalism used in the theoretical modelling.

## 2. Materials and Methods

### 2.1. Experimental Setup

For the collection of *xilematic* bioelectric potentials from one tree, ten stainless steel (AISI 316) threaded rods of 6 mm diameter were utilised. These rods were evenly spaced along the trunk, uncovered portion of roots and logs, with a separation distance of 50 cm. Each electrode pair was connected to a differential amplifier prior to recording the data.

For the collection of *floematic* bioelectric potentials, two recording circles were positioned around the trunk of another tree. One circle was located 1 m above the ground level with a radial distance of $60^{\circ }$, while the other was located 3 m above the ground level with a radial distance of $90^{\circ }$ (see Figure 1). As with the *xilematic* potential collection, each electrode pair was connected to a differential amplifier prior to analogue to digital conversion.

The same threaded rods were used to collect bioelectric signals from five logs. The first rod was inserted at the top of each log, and the second was positioned along one of the roots, 50 cm away from the first. Both rods were then connected to the differential amplifier. Signals were transmitted using the double-shielded ultra-low resistance INCA1050HPLC cable from MD Italy, designed for high-fidelity audio applications. DI-710-US data loggers were used in differential mode, recording data at a frequency of 1 sample per second per channel with 16-bit resolution and a full range of 1 V.

The acquired data were smoothed using a first-order Savitzky–Golay function with a variable-size window ranging from 41 to 301 points based on the noise level. The electrodes were labelled with the Greek letters α,β,γ,δ,ϵ,η,ζ,θ,ι, and κ. The collection sites were labelled with the Latin letters ‘B’, ‘E’, and ‘G’.

Collection site ‘B’ represents a healthy, sun-exposed tree of approximately 70 years old, site ‘E’ represents a portion of five dead logs in an area devastated by the 2018 Vaia storm, and site ‘G’ represents a healthy, shadowed tree of approximately 20 years old. These collection sites were selected to showcase the varying ages and vitality levels that can be found in forests. The thermographies of trees at sites ‘B’ and ‘G’ and one log at site ‘E’ were captured at 30 minute intervals from a fixed orientation using a FLIR camera with a 240×240 pixel CCD sensor. The frames were then cropped and masked to enhance the signal-to-noise ratio of the relevant portions of the trunks.

The images were analysed using MATLAB R2021b, resulting in the computation of Shannon entropy for each frame plotted over time. By aligning the heterogeneous datasets of Shannon entropy and bioelectric potentials on a common time scale, a direct correlation was established between them. Further numerical techniques were utilised to extract complexity metrics from the biopotentials as detailed in Section 2.2. Our research focused on comparing the Shannon entropy and fractal dimensions as this comparison could provide valuable insights into the sensory capabilities of living plant ecosystems.

### 2.2. Numerical Analysis

In various demanding signal-processing systems and applications, using second-order statistics, such as the mean, variance, and correlation, becomes insufficient when the data deviate from a Gaussian distribution and the adaptive system is nonlinear [37,38,39,40]. In such scenarios, higher-order statistics are necessary to represent the characteristics of linear/nonlinear adaptive signal-processing systems with greater accuracy. Furthermore, the concept of information theory suggests that the level of uncertainty determines the information value of data. Indeed, the data carry little information if an event is highly probable. By using metrics, such as the entropy, Simpson diversity, expressiveness, and Lempel–Ziv complexity, higher-order statistics help to reduce uncertainty.

In this study, we computed the following information–theoretic complexity measures to characterize the spatiotemporal activity patterns in raw data with a time window of 30 min, making sure to link the complexity calculation with the instantaneous thermography of each particular site.
(1)The Shannon entropy ($I_{1}$) is a measure of the uncertainty of a discrete random variable. Given a random variable *s* with *n* elements, $s=\{s_{1},s_{2},\cdots ,s_{n}\}$, and its probability distribution $p\left(s\right)=\{p\left(s_{1}\right),p\left(s_{2}\right),\cdots ,p\left(s_{n}\right)\}$, the Shannon entropy can be expressed mathematically as in Equation (Equation 1).
(1)$$I_{1}=-\sum_{i=1}^{n}p\left(s_{i}\right)\log\left(p(s_{i})\right).$$(2)The Rényi entropy ($I_{2}$) constitutes a fundamental aspect of the notion of generalised dimensions, which play a crucial role in statistics as a measure of diversity [41]. The mathematical formulation of this entropy is given by Equation (Equation 2). In the present study, we fixed the parameter *q* to 2.
(2)$$I_{2}=\frac{1}{1-q}\left(\ln\left(\sum_{i=1}^{n}p{\left(s_{i}\right)}^{q}\right)\right)$$(3)The Tsallis entropy ($I_{3}$), a generalization of the standard Boltzmann–Gibbs entropy, represents a non-extensive entropy measure [42]. The mathematical expression of this entropy is represented by Equation (Equation 3), in which *q* and *k* denote the degree of non-extensivity and a positive constant, respectively. Our empirical study implements a setting of q=2 and k=1.
(3)$$I_{3}=\frac{k}{1-q}\left(1-\sum_{i=1}^{n}p{\left(s_{i}\right)}^{q}\right)$$(4)The space filling ($I_{4}$) is defined as the ratio of non-zero elements in the signal *s* to the total length of the signal.(5)The Expressiveness ($I_{5}$) is determined as the ratio of the Shannon entropy ($I_{1}$) to the space filling ($I_{4}$). This metric provides an indication of the “economy of diversity”.(6)The diversity index ($I_{6}$) quantitatively assesses the number of unique activities of interest present in the acquired signal and considers phylogenetic relationships between these activities, including aspects, such as the richness, divergence, and evenness. Its mathematical formulation is given by Equation (Equation 4), where we set the parameter *q* to 3.
(4)$$I_{6}=\frac{1}{\sqrt[{q-1}]{\sum_{i=1}^{n}p\left(s_{i}\right)p{\left(s_{i}\right)}^{q-1}}}={\left(\sum_{i=1}^{n}p{\left(s_{i}\right)}^{q}\right)}^{1/1-q}$$(7)The Simpson diversity ($I_{7}$) is a measure of the concentration of individuals classified into types and can be calculated as $I_{7}=\sum_{i=1}^{n}p{\left(s_{i}\right)}^{2}$. The range of $I_{7}$ is between 0 and 1, with 1 indicating infinite diversity, and 0 representing no diversity.(8)The Lempel–Ziv complexity ($I_{8}$) is utilised as a measure of temporal signal diversity, with a focus on its compressibility. The calculation of $I_{8}$ was performed using the Kolmogorov complexity algorithm [43]. This metric has been found to be valuable in providing a scalar measurement for estimating the bandwidth of random processes and quantifying the harmonic variability in quasi-periodic signals.(9)The perturbation complexity index (PCI) ($I_{9}$) is determined by the normalisation of the Lempel–Ziv complexity of the spatiotemporal pattern of the signal, relative to its Shannon entropy ($I_{1}$).(10)The Kolmogorov complexity ($I_{10}$) is a metric that evaluates the information content of a signal. It quantifies the minimum length of a binary code required to describe the signal, assuming an optimal encoding, and can be expressed mathematically as per Equation (Equation 5).
(5)$$I_{10}=\min\{|P||P\in {0,1}^{*},U\left(P\right)=s\}$$
where U(P) is the output of the binary program P when executed on a fixed reference universal Turing machine, and |P| is the length of the binary program P.(11)The fractal dimension ($I_{11}$) quantifies the self-similarity of a signal across multiple scales. In accordance with the notion that signals exhibiting more irregular or chaotic patterns are expected to exhibit a higher fractal dimension than signals featuring more regular or predictable patterns, in this study, the fractal dimension of a signal was calculated using the Higuchi method [44]. This method is based on the fractal dimension formula for a curve as represented by $I_{11}=\frac{\log(n)}{\log(1/d)}$. Here, the parameter *d* represents the step size used in signal sampling, and log(·) represents the natural logarithm.

## 3. Results

The response data collected from the five differential channels of each study site were found to be inherently noisy.

In Figure 2, Figure 3 and Figure 4 (plots (a)), we present a series of plots with 20,000 s of continuous recordings from a healthy old sun-exposed tree at site ‘B’, five dead logs at site ‘E’, and a healthy young shadowed tree at site ‘G’, respectively. The underlying oscillations and correlations between channels become more noticeable (as shown in plots (b) of each Figure) when the time scale is reduced to 1 h and when the first-order Savitzky–Golay smoothing function with 101 points is used.

In evaluating differences in biopotential values, it is essential to consider the relative scale where the difference is always between two electrodes. For vertically aligned electrodes at site ‘B’, the potential difference for the α−β and ι−κ pairs is around 100 to 120 mV, whereas for the γ−δ and ϵ−ζ pairs, it ranges from 20 to 40 mV. This slight difference between the two electrode groups may be attributed to the contact resistance at the collection site, which is beyond the study’s control. It is possible that the natural production of resin could increase the contact resistance and diminish the magnitude of the measured signals. Interestingly, the η−θ pair exhibited the highest potential difference, at approximately 650 mV, as the piezometric pressure of water in the soil generated an ionic gradient whose boundary occurred at a specific height, corresponding to the electrode level.

At site ‘E’, potential differences range from 70 to 80, 120 to 140, 20 to 30, 10 to 20, and 70 to 90 mV for dead logs, with a lower dispersion of values. For the electrodes positioned radially around the trunk at site ‘G’, the higher circle has potential differences of about 80 to 100 mV, while the lower circle features potential differences ranging from 160 to 200 mV, except for the ι−κ pair, which has values of 900 to 930 mV. This high discrepancy is attributed to the position of the electrodes, which face the sun during recordings and feature the highest potential difference, demonstrating that sensors can accurately trace the plant’s physiological activity. In addition, observations on the signal amplitudes, frequencies, and noise are reflected in the complexity measures.

In the *xilematic* recordings, since lymph movements along the trunk are linked to ionic conduction, temporal shifts between channels can be easily justified when a particular electrical feature, such as a valley or a peak, propagates.

In the *floematic* recordings, correlations can also be observed, particularly between the two differential channels operating across the sun-exposed coordinates observable between east and west passing from the south. However, the dead logs feature a less correlated structure, with some exceptions possibly due to anastomosis of roots where electrical connections may play a role in correlating signals. It is noteworthy that the *floematic* recordings of the young tree exhibit a more pronounced bioelectric activity as demonstrated by the multiple jumps and non-steady potentials measured. The corresponding complexity measures, specifically the Kolmogorov complexity (as shown in plots (c)) and the fractal dimension (as shown in plots (d)), display a greater extent and more significant variation compared to the other two cases.

The complexity measures were computed using the formulas described in Section 2.2 from the raw signals. Each differential channel generated a complexity curve as illustrated in panels (c) and (d) of Figure 2, Figure 3 and Figure 4. The samplings caused spontaneous activity that induced abrupt jumps in the biopotential recordings. This effect was particularly noticeable in the ϵ−ζ couple shown in Figure 4a. However, the older tree (Figure 2a) and logs (Figure 3a) exhibited more uniform noise without such jumps.

We employed the Kolmogorov and other complexity measures to provide a more “holistic” interpretation of the biopotential curves. A higher complexity value indicated that there were more biopotential fluctuations per time unit. The number and amplitude of fluctuations varied across channels depending on the underlying physiological activity that generated the signal. Electrical probes served as point sensors that recorded phenomena related to living beings with vast complexity. Differences in entropy were also impacted by photosynthetic activity, which was more influenced by topmost electrodes or couples situated closer to the Sun position.

For example, in all complexity measurements for site ’B,’ we observed that the complexity was maximal for the differential couple alpha−beta that corresponds to the topmost electrodes positioned at 3 and 2.5 m from the ground level. On the other hand, the differential couple corresponding to the trunk basement, η−θ, exhibited higher complexity discrepancy and was, thus, excluded from the average calculation but was retained for thermographic correlation analysis.

For site ‘G’, we observed that the complexity of the two differential couples positioned 3 m above the ground level was lower and well-separated from the three couples positioned 1 m above the ground level. Despite a similar trend, calculating the average for this case is appropriate and does not pose any issues for the subsequent analysis.

Our study utilised thermographic imaging and bioelectric potential recordings at three sites as shown in Figure 5. The data was synchronised in time, and complexity analysis was performed to produce comparable results. After registering and masking each frame, the Shannon entropy was calculated and compared with the fractal dimension of electrical signals as shown in Figure 6.

The results reveal a strong correlation between thermodynamics and bioelectricity at sites ‘B’ and ‘G’ as demonstrated by the overlap of the two curves. At site ‘B’, the couple η−θ corresponding to the trunk basement (depicted in the top row plots of Figure 5) displayed a significant correlation. The young tree at site ‘G’ displayed an even more pronounced correlation as it exhibited a variety of complex phenomena with a peculiar trend of complexity over time.

In contrast, the correlation between thermodynamics and bioelectricity was weak in the case of dead logs due to less spontaneous electrical activity and poor synchronisation between differential couples connected to different individuals. However, our findings demonstrate that the average fractal dimension of bioelectric potentials is directly related to the Shannon entropy of thermographic images.

The bioelectric potential curves recorded at different sites can be compared by calculating their quadratic cross-correlations to determine if a time lag leads to better synchronisation of the signals. This analysis was performed by dividing the data into 1 h segments, selecting one signal as a reference, and shifting another signal in time relative to the reference, both from tail-to-head and head-to-tail. At each step, the quadratic cross-correlation was calculated. As shown in Table 1, several regimes were observed over 24 h, with most cases showing a perfect synchronisation with a time lag of less than a second or slightly larger in the evening and, in some cases, a perfect synchronisation in the morning and afternoon.

In the subsequent Section 4, we explore the possibility of explaining the observed outcome. In one instance (late at night), the time lag corresponded to a considerably slower propagation speed of approximately 1 cm/s, which is consistent with previous observations of the physical transport of signals by the lymph [32]. Figure 7 presents the visual representation of the quadratic cross-correlation functions, particularly in the case of chemical diffusion and long lags, which demonstrate that their structure is highly similar and exhibits two correlations.

One occurs around 1100 s, corresponding to a propagation speed of 1.5 cm/s, and the other, less prominent, occurs at the double the timescale, 6.4 mm/s. This could be attributed to the presence of multiple ionic species, each contributing to the propagation of signals between distant individuals through the root network and their differing diffusion speeds. We can conclude that bioelectrical signals from various trees may display accurate synchronisation or temporal deviation, depending on the hour of the day and the metabolic condition of the forest. These factors are contingent upon prevailing environmental conditions, such as temperature, humidity, insolation, and water availability.

## 4. Theoretical Modelling

Living plants are dynamic systems that constantly exchange matter and energy with their environment in various forms. Consequently, they are not “closed” systems but rather dissipative systems that “sense” the environment through the inflow and outflow of matter and energy. In the previous sections, the reported experimental results confirmed the permanent “dialogue” of the trees (and logs) with the environment in which they are embedded as well as a dialogue among the trees themselves.

In this respect, the results reported, for example, in Table 1 and Figure 7, suggest that there might be two main characterisations of the said dialogue. From one side, communication rests on the exchange of chemical vectors travelling at different speeds through the roots, soil, and the atmosphere, in general. On the other hand, the cases of the observed smallness of time-scale synchronisation are difficult to explain in classical terms of chemical or other kinds of mediators whose speed would be exceedingly high, suggesting that the underlying molecular dynamics of the trees might allow phase-correlations (entanglement), by its nature—not mediated by messengers—but emerging from the plants sharing the same environmental background (the same ground state in the language of our modelling; see below).

This, in turn, suggests investigating quantum field theory (QFT) dynamics resulting in long-range correlations at the molecular level. The theoretical modelling proposed in this section is motivated by, and moves along the view derived from, the presented results and provides the QFT formalism accounting for them. Our modelling framework is mainly targeted towards comprehending and elucidating specific findings presented in Section 3, which were also illustrated in Figure 2, Figure 3, Figure 4, Figure 5, Figure 6 and Figure 7 and summarised in Table 1.

Specifically, we aim to shed light on the following aspects: (1) the diverse array of dynamic regimes observed in the recorded bioelectric potential data, which provide insights into the physiological activity of trees; (2) the shift in scale from microscopic events involving electrons, ions, and molecules to macroscopic manifestation in terms of bioelectric potential recordings; (3) the pivotal role played by temperature and entropy in mediating the transition between different dynamical regimes; (4) the manner in which the fractal dimension, its changes, and its relationship with temperature and entropy contribute to the characterisation of the recorded bioelectric potential data; and (5) the observations on lag time scales of biopotential fluctuations at various sites, which suggest inter-tree communication facilitated by the exchange of chemical and other physical vectors, and phase-correlations through the shared environment in which trees exist.

The present study offers a qualitative comprehension of the experimental findings using a modelling approach that integrates fundamental first principles into a cohesive theoretical framework. Despite its limitations, our model’s strength lies in its unifying ability to reconcile the observations with the underlying theoretical principles. As future prospects, we intend to pursue a more comprehensive and descriptive modelling approach that would yield quantitative fitting of the presented results.

The subsequent part of this section is arranged in the following manner. First, we elaborate in Section 4.1 on how our theoretical model achieves the goal of elucidating the outcomes of points (1) and (2). In addition, we evaluate points (3) and (4) in Section 4.2 and address points (1)–(3) and (5) in Section 4.3. Finally, comprehensive formal details are presented in Section A.1, Section A.2 and Section A.3.

### 4.1. On the Molecular Dynamics Underlying the Observed Electrical Activity

Previous sections demonstrated that the molecular activity in living plants is impacted by a range of endogenous and exogenous factors, including pressure agents, thermal and chemical gradients, light and other electromagnetic (EM) radiation, and gravitational forces from the earth, moon, and sun. One significant consequence of these influences is the breakdown of molecular dynamics symmetries.

In this section, we mainly focus on the dynamics of water molecules, which play a central role in the lives of plants, and other molecules that have specific electrical dipole moments due to the charge configuration of their electronic shells. Since these molecular dipoles may be, in principle, oriented in any direction, their dynamics are symmetric under spherical rotational transformations. However, the aforementioned forces and agents cause this symmetry to break down, with dynamic consequences that can be well-understood within the framework of QFT. In the phenomenon of spontaneous breakdown of symmetry (SBS), the symmetry-breaking agents alter the system state; however, they do not modify the field evolution equations.

This contrasts with explicit symmetry breakdown, where the breaking agents modify the field equations [45,46,47,48]. SBS is known to generate a network of long-range correlations among the system constituents in their lowest energy state (the vacuum or ground state of the system). This leads, thus, to the formation of ordered patterns as stated by the Goldstone theorem in QFT. These correlations, characterised by in-phase, non-destructive interference, are referred to as *coherent* correlations. They are responsible for the system’s ordering and for its macroscopic functional activity (such as in a crystal or a magnet).

The “order parameter” provides a measure of the degree of order in the system’s ground state; it is a classical field, whose observed values do not, in fact, depend on quantum fluctuations. For instance, in magnets and in crystals, the order parameter is the magnetisation and density, respectively. In the case of molecular electrical dipoles, the order parameter is the non-zero polarisation density P(x,t), which would be zero on average in the absence of SBS.

In QFT theories with SBS, the boson quanta associated in the infinite volume limit with correlation waves are the massless Nambu–Goldstone (NG) modes or quanta [49,50,51]. In the case of dipole correlation waves, these NG modes are called dipole-wave quanta (DWQ) [52,53]. The system’s ground state is characterised as a *coherent* condensate of DWQ.

Endogenous or exogenous factors may induce variations in the value of the order parameter, corresponding to changes in the condensation density and, therefore, to transitions of the system through different dynamic regimes—referred to as *phase transitions*. This is a characteristics of living systems, which undergo a continuous process of phase transitions.

Already at this point of our discussion, we see that the theoretical modelling includes the transition from the microscopic scale of the molecular activity to the macroscopic scale of the classical order parameter. We will return to this point in the following. Moreover, the description of the tree system living over a manifold of different dynamical regimes, continuously undergoing through phase transitions is also a specific feature of our QFT theoretical frame. This point will be further discussed in the following.

Consequently, the connection between the condensation of DWQ and the electrical pulse activity discussed in Section 3 becomes crucial. To illustrate this connection, we simplify the analysis by limiting ourselves only to the space components of the EM vector potential A(x,t). Consider the (local) U(1) gauge transformation: (6)A(x,t)→A(x,t)+∇λ(x,t),
and adopt, as customary, the Coulomb gauge condition ∇·A(x,t)=0. Consistently with this condition, the constraint $\nabla^{2}\lambda \left(x,t\right)=0$ is imposed. The SBS of the global U(1) symmetry results in the non-vanishing value of the order parameter v(x,t), which is found to be ${\left|v\left(x,t\right)\right|}^{2}=2P\left(x,t\right)$ [48,53].

Let the NG boson, originating from the U(1) SBS [47,51,53], be represented by the real field χ(x,t). It can be shown that the condensation of the χ(x,t) field in the system ground state, described by the transformation χ(x,t)→χ(x,t)+(q/ℏc)λ(x,t), induces the gauge transformation (Equation 6), provided that the consistency conditions $\nabla^{2}\chi \left(x,t\right)=0$ is also satisfied (few formal details in the Section A.1). We notice that observable effects only appear when λ(x,t) is a singular and/or topologically non-trivial function (e.g., in the case of vorticity or ring-like structures) [46,48,51,54].

In these cases, different paths between two points (two electrodes) *A* and *B* are not equivalent, which underlies the high complexity detected in data acquisition. In conclusion, our analysis, which is consistent with the established results in QFT [47,48,51,54], shows that the evolution of the classical vector potential is directly linked to the topologically non-trivial dynamics of coherent boson condensation in SBS.

We will see in the following that the fractal dimension observed in the measurements can be also described in terms of the coherent state dynamics of the molecular activity.

### 4.2. Thermal Effects, Free Energy, Entropy, and Fractal Self-Similarity

The results described in Section 3 show that temperature and its changes play a relevant role in the bioactivity of plants. In each one of the dynamical regimes through which they evolve (the *phase transitions*), considered as equilibrium (or quasi-equilibrium) states for the system coupled to the environment at a given temperature *T*, the free energy F=U−TS has to be minimized according to the general laws of thermodynamics, namely dF=dU−TdS=0, with *U* denoting the internal energy, *S* the entropy, and the Boltzmann constant $k_{B}$ set equal to 1 for simplicity.

In particular, the lesson coming from the results presented in Section 3 is that a sort of “internal degree of freedom” exists for the living plants, such that energy transmutation is possible from ordered configurations (low entropy) to (random) kinetic configurations or vice versa, TS↔U. The theoretical modelling must, therefore, account for such a feature emerging from the observations.

Let us start by remarking that, according to our theoretical framework, the behaviour of the boson DWQ coherent condensate can be well-approximated as that of an ideal gas due to its coherence. Due to impurities or boundary effects, DWQ develops a non-vanishing effective mass, $m_{eff}$ [53]. In accordance with the equipartition theorem, the magnitude of the momentum p is related to the temperature *T* and the effective mass $m_{eff}$ through $p^{2}/\left(2m_{eff}\right)=\left(3/2\right)k_{B}T$.

The correlation wavelength λ is related to the momentum *p* through the de Broglie relation λ=h/p, with *h* being the Planck constant. The finite size over which the coherent correlated domain extends can be expressed as $R=\hbar /(m_{eff}\;c)=n\lambda /2$, where *c* is the speed of light, ℏ=h/(2π), and *n* is an integer number. As established in previous studies [32,53], in combining these relations, one finds
(7)$$T=\frac{\pi hc\;n^{2}}{6k_{B}R}.$$
The validity of the said approximation is confirmed by the fact that the inverse proportionality expressed by Equation (Equation 7) between *T* and the range *R* over which the coherent correlation extends is indeed consistent with the law of the minimization of the free energy dF=0, i.e., dU=TdS. This can be derived from (Equation 7) [32] by differentiating both sides, obtaining
(8)$$dT=\frac{T}{C_{V}}dS,$$
with $C_{V}\equiv {\left(\frac{dU}{dT}\right)}_{V}={\left(\frac{dQ}{dT}\right)}_{V}$ the specific heat at constant volume *V* of the DWQ ideal gas and $dS=dQ/T=C_{V}d\left(logT\right)$. Considering that at constant *V*, $dU=C_{V}dT$, (Equation 8) gives dU=TdS.

The analysis of the thermographic pictures (Figure 5) presented in Section 3 suggests that a dynamical control mechanism is activated in the trees, aimed at keeping their temperature higher than the environment’s low temperature. Suppose that a decrease (increase) in the environmental temperature needs to be contrasted by the tree for its survival. Equation (Equation 7) shows that, in order to increase (decrease) *T*, the correlation range *R* must be accordingly decreased (increased), with corresponding disordering and entropy increase (ordering and entropy decrease).

The tree system, thus, reacts to the external perturbing actions according to the Le Chatelier principle [55,56] by ‘disinvesting’ energy (entropy increase), i.e., releasing the energy stored in ordered patterns to the gas-like kinetic regime or harvesting energy from the gas-like kinetic regime and ‘investing’ it into the formation of coherent in-phase correlated patterns (entropy decrease and coherent boson condensation) with all processes constrained by the dF=0 condition. We thus see that the mentioned energy transmutation from ordered configurations to kinetic ones or vice versa, TS↔U, is possible.

This leads us to the further possibility offered by our theoretical modelling—namely, of understanding the dynamic origin of the observed fractal dimension of bioelectric potential, of its changes, and its relation with entropy (cf. Figure 2d, Figure 3d, Figure 4d and Figure 6).

It has been shown [48,57,58] that an isomorphism exists between fractal self-similarity and coherent states in QFT, and therefore, between the fractal dimension and the entropy associated with the coherent condensate (cf. Section A.2). In log–log plots of conjugate variables (e.g., the logarithm of power spectral density versus the logarithm of frequency), if fractal self-similarity is present in the phenomenon under study, the measured data follow, in their log–log plot, a distribution along a straight line, whose slope is related to the fractal dimension. In order to make the reading easier, we report in the Section A.2 some details of the related mathematical formalism.

Here, we limit ourselves to note that, not only is the observed fractal behaviour described in our scheme but also the changes of the fractal dimension reported in Figure 2d, Figure 3d, Figure 4d and Figure 6 are understood. In fact, from the discussion above presented, at constant *V* and constant (or quasi-constant) *T*, the system may move along the straight line of slope T=dU/dS in the plane (U,S). It is in such processes of changes in the coherent condensate densities (described by the corresponding changes in the entropy) that the observed changes in fractal dimensions occur (in agreement with the intrinsic relation between the fractal dimension and entropy in our modelling). These features also emerge in the following discussion on the system QFT coherent state.

### 4.3. Entanglement and Collective Dynamical Effects

As already observed, living plants are open, dissipative systems, and the QFT canonical formalism requires that the description of their states also include the effects of temperature and the environment [46,47,48,59]. Since fluxes ingoing to the system are outgoing from the environment, and vice versa, the last one is described as the time-reversed image of the system. Then, the finite temperature QFT state describing the system and its environment is [32,59].
(9)$$|0\left(\theta \left(\beta \right)\right)\rangle =\prod_{k}\frac{1}{\cosh\theta_{k}\left(\beta \right)}\exp\left(\tanh\theta_{k}\left(\beta \right)\;a_{k}^{†}{\tilde{a}}_{k}^{†}\right)|0\rangle ,$$
which is the SU(1,1) generalised coherent state [60] at finite temperature *T*. Moreover, as discussed below and in the Section A.2 and Section A.3, it is found to be an entangled and squeezed state. For simplicity, the time-dependence of β(t) is not shown, $\beta \left(t\right)\equiv \beta =1/k_{B}T$. The state is normalized to 1, 〈0(θ(β))|0(θ(β))〉=1,∀θ(β),∀β,∀t, and the operators ${a^{†}}_{k}$ and $a_{k}$ are the creation and annihilation operators, respectively, entering in the expansion of the NG correlation field χ(x,t). The operators ${\tilde{a}}_{k}^{†}$ and ${\tilde{a}}_{k}$ are the creation and annihilation operators representing the environment (the thermal bath) [46,47,48,59].

Equation (Equation 12) shows that changes in temperature, $\beta \to \beta^{\prime }$, lead to different dynamical regimes in the infinite volume limit. This is also formally described by the change in the slope of the straight line in the (U,T) plane, namely of the fractal dimension (cf. Section 4.2 and Section A.3) characterising the system dynamics, a new ratio dU/dS between internal (kinetic) energy and energy stored in the dipole wave ordering correlations.

We introduce in the Section A.1, Equation (Equation 13), the entropy $S_{a/\tilde{a}}$ and the free energy $F_{a/\tilde{a}}$ for the modes $a_{k}$ and ${\tilde{a}}_{k}$. From Equation (Equation 17), we see that, for $dF_{a}=dU_{a}-\left(1/\beta \right)dS_{a}=0\;$, variations in time of $N_{a_{k}}\left(\theta \left(\beta \left(t\right)\right)\right)$ (the number of $a_{k}$ modes condensed in |0(θ(β))〉, i.e., of the dipole wave ordering correlations), produce equivalent changes in the “internal energy” $dU_{a}$ and in entropy $dS_{a}$. There is a redistribution of the energy from the ordered structure to the kinetic energy sector or vice versa, thereby, confirming our discussion in Section 4.2.

Variations in the entropy, related to the Kolmogorov entropy in nonlinear dynamical systems [61] are induced by fluctuations for small $\delta \theta_{k}$ and negligible $\partial \delta \theta_{k}/\partial t$ at the minimum condition $dF_{a}=0$ (cf. Equation (Equation 19)). Phase correlation in the state |0(θ(β))〉 is manifest since it is an entangled state.

Remarkably, such an entanglement has effects at the macroscopic level since, as seen, the em equations at a classical level are controlled by the entangled coherent boson condensate [47,48,51,54]. On the other hand, the QFT state |0(θ(β))〉 determines measurable quantities, such as thermal averages and the entropy, and is, therefore, called a “macroscopic quantum state”, which is realistic in the manifestations of its coherent condensate, e.g., the condensate in a magnet or a crystal.

Moreover, the extremely short time lags (cf. Figure 7 and Table 1) do not imply a violation of relativity postulates since the phase velocity is not bounded by the light velocity. The entanglement appears, thus, not as an effect of spooky forces at a distance [62,63,64] but of the embedding of the plants in the common environmental bath, of which themselves are part, described by the state |0(θ(β))〉. The covariance $cov\left(N_{a},N_{\tilde{a}}\right)\equiv \langle N_{a}N_{\tilde{a}}\rangle -\langle N_{a}\rangle \;\langle N_{\tilde{a}}\rangle$ gives a measure of the degree of entanglement.

One finds $cov\left(N_{a},N_{\tilde{a}}\right)=\left(1/4\right)\sinh^{2}\theta \neq 0$ for θ≠0. In contrast, $cov(N_{a},N_{\tilde{a}})=0$ for non-correlated modes since, in that case, $\langle N_{a}N_{\tilde{a}}\rangle =\langle N_{a}\rangle \;\langle N_{\tilde{a}}\rangle$ (expectation values in |0(θ(β))〉 are denoted by the symbol 〈**〉, and the subscript k has been omitted for simplicity). For non-zero values of $\langle {\left(\Delta N_{a}\right)}^{2}\rangle$ and $\langle {\left(\Delta N_{\tilde{a}}\right)}^{2}\rangle$, one might also normalize the covariance dividing it by ${\left(\langle {\left(\Delta N_{a}\right)}^{2}\rangle \right)}^{1/2}\;{\left(\langle {\left(\Delta N_{\tilde{a}}\right)}^{2}\rangle \right)}^{1/2}$, obtaining then $J(N_{a},N_{\tilde{a}})=1$ for the linear correlation coefficient *J* [32,62,63,64,65].

As a final result, as already observed in [32], the forest appears to be an *in-phase collective dynamical system*.

## 5. Conclusions and Future Prospects

In this study, we aimed to demonstrate the correlation between electrical activity patterns of superior plants and thermographic images in a *Picea abies* forest. The experiments were conducted in several sites with different stages of plant development, including the dead state, and the data collected were time-stamped precisely to ensure accuracy, thus, providing a unique dataset to analyse.

To analyse the data, we employed higher-order signal statistics and complexity metrics, which enabled us to highlight the significant correlation between bioelectric signals and thermal data. The bioelectric signals were found to reflect the metabolic activity of the plants, while the thermal data reflected their thermodynamic state.

A theoretical analysis based on quantum field theory was used to explain the relationship between the two aspects, and we viewed the forest as a collective array of in-phase elements that are naturally correlated and tuned depending on environmental conditions. Further analysis of the lag time between simultaneous recordings from different locations showed that the synchronisation of bioelectric signals featured at least two regimes: one with an exact synchronisation and one with a much higher time lag, possibly due to physical diffusion.

In conclusion, this study provides valuable insights into the correlation between electrical activity and thermographic images in a forest. Furthermore, our findings open up roads for future research to explore the impacts of environmental changes on the bioelectric response and the predictability of such changes.

## Figures and Tables

**Figure 1 biomimetics-08-00122-f001:**
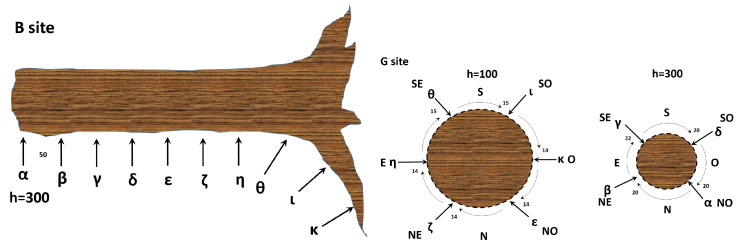
**Left**: a schematic illustration of the *xilematic* bioelectric potential recording of site B from a healthy tree (rotated 90° counterclockwise). **Right**: the *floematic* bioelectric potential recording of site G from a healthy tree. Here, we show two sections at different heights from the ground. The numbers indicate distances in centimetres, with relevant coordinates specified as N = North, S = South, E = East, W = West, NE = North-East, SE = South-East, NW = North-West, and SW = South-West.

**Figure 2 biomimetics-08-00122-f002:**
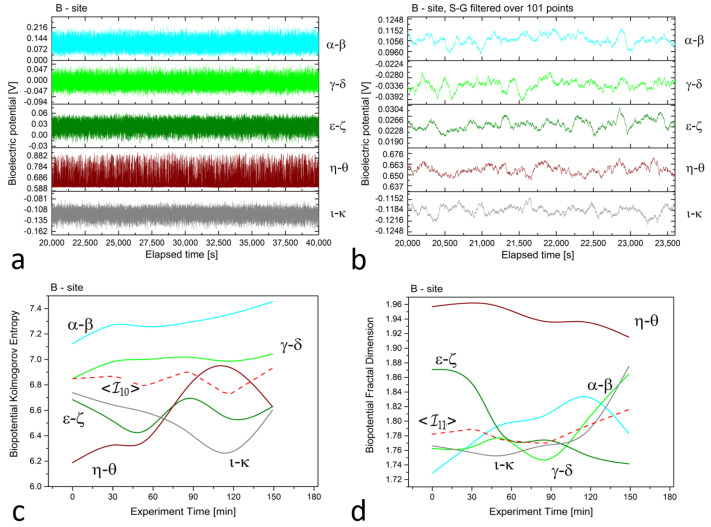
(**a**) Raw recordings from five differential *xilematic* channels situated along the height of an old, healthy tree, captured over a 20,000 second continuous recording. (**b**) The plot of 1 h of data after applying the first-order Savitzky–Golay smoothing function with a window of 101 points. (**c**) The Kolmogorov complexity of the bioelectric potential data collected during thermographic data acquisition. All channels are presented along with their average (the dashed red curve). (**d**) The fractal dimension of the bioelectric potential data collected during thermographic data acquisition. All channels are presented along with their average (the dashed red curve).

**Figure 3 biomimetics-08-00122-f003:**
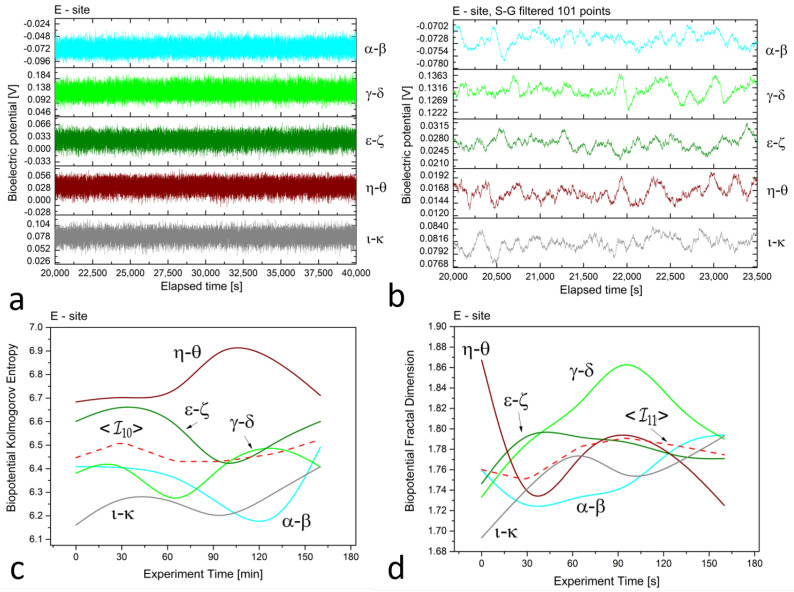
(**a**) Raw recordings from five differential *xilematic* channels positioned on five different dead logs in proximity, captured over a 20,000 second continuous recording. (**b**) The plot of 1 h of data after applying the first-order Savitzky–Golay smoothing function with a window of 101 points. (**c**) The Kolmogorov complexity of the bioelectric potential data collected during thermographic data acquisition. All channels are presented along with their average (the dashed red curve). (**d**) The fractal dimension of the bioelectric potential data collected during thermographic data acquisition. All channels are presented along with their average (the dashed red curve).

**Figure 4 biomimetics-08-00122-f004:**
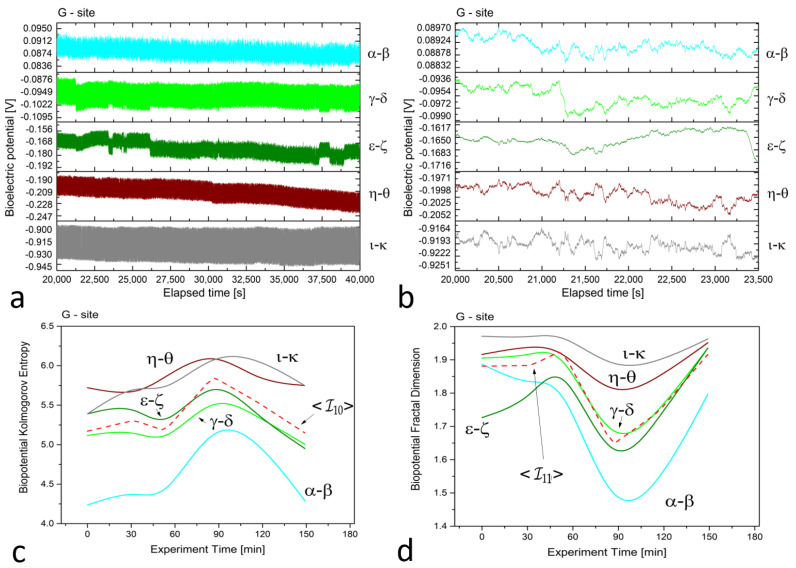
(**a**) Raw recordings from five differential *floematic* channels situated along the trunk of a young, healthy tree, captured over a 20,000 second continuous recording. (**b**) The plot of 1 h of data after applying the first-order Savitzky–Golay smoothing function with a window of 101 points. (**c**) The Kolmogorov complexity of the bioelectric potential data collected during thermographic data acquisition. All channels are presented along with their average (the dashed red curve). (**d**) The fractal dimension of the bioelectric potential data collected during thermographic data acquisition. All channels are presented along with their average (the dashed red curve).

**Figure 5 biomimetics-08-00122-f005:**
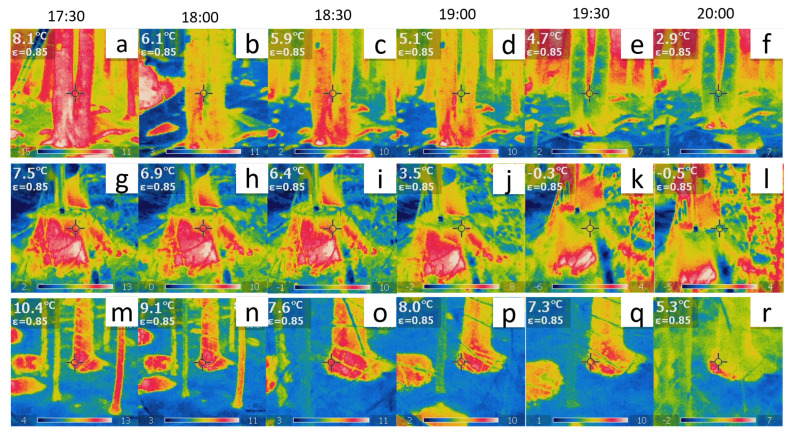
(**a**–**f**) Thermographic images of the tree at Site ‘B’. (**g**–**l**) Thermographic images of one of the five logs at Site ‘E’. (**m**–**r**) Thermographic images of the tree at Site ‘G’. The temperature readings in the top-left inset of each plot are direct observations with a fixed emissivity from the pixel indicated by the middle cross. The actual time scale is also indicated.

**Figure 6 biomimetics-08-00122-f006:**
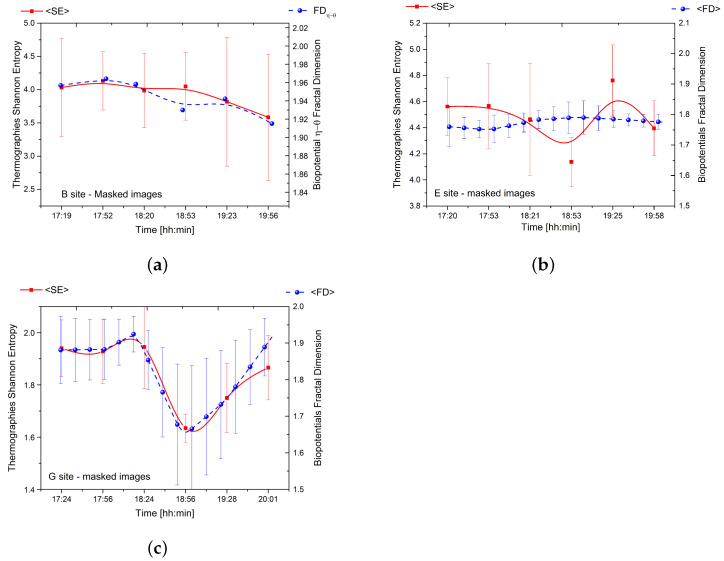
(**a**) The relationship between the average Shannon entropy of thermographies and the fractal dimension of couple η−θ for the tree located in site ‘B’. (**b**) The correlation between the average Shannon entropy of thermographies and the average fractal dimension of bioelectric potential for one of the five logs located in site ‘E’. (**c**) The relationship between the average Shannon entropy of thermographies and the average fractal dimension of bioelectric potential for the tree located in site ‘G’.

**Figure 7 biomimetics-08-00122-f007:**
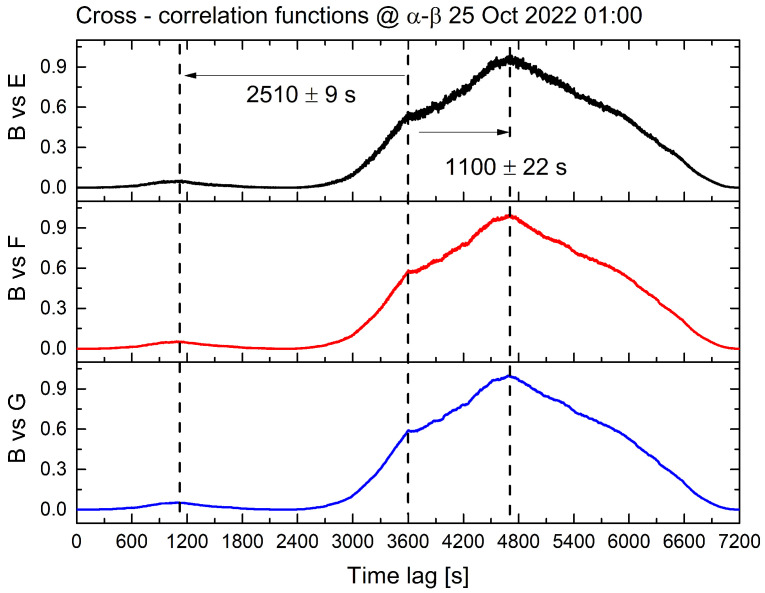
The results of a quadratic cross-correlation analysis on biopotential signals collected from all sites, with a data chunk of 1 h. The biopotential vectors from site ‘B’ were compared to those from sites ‘E’, ‘F’, and ‘G’ to investigate temporal correlations. The cross-correlation functions for each comparison are represented by the black curve (site ‘B’ vs. site ‘E’), the red curve (site ‘B’ vs. site ‘F’), and the blue curve (site ‘B’ vs. site ‘G’), respectively. The similar shapes of the curves demonstrate the similarity in the behaviour of biopotential fluctuations across all comparisons. The time lag at which the cross-correlation reaches its absolute maximum is approximately 1100 s, and a relative maximum occurs at approximately double the time lag. The compared sites are located at about the same distance, approximately 16 m.

**Table 1 biomimetics-08-00122-t001:** Analysis of the synchronisation time scales of biopotential fluctuations at different sites through quadratic cross-correlation. This analysis was conducted by shifting the biopotential vectors of channels α−β and computing the quadratic cross-correlation function for all lags within a range of one-hour delay or anticipation. The values in the table represent the average of the calculations performed on all permutations of the sites ‘B’, ‘E’, ‘F’, and ‘G’. The average propagation speeds were calculated based on the average linear ground distance between the sites, which was 16 m.

Date	Time	Average Air Temperature [°C]	Average Lag Time [s]	Average Speed [m/s]
24 October 2022	19:00	8.1	0.5	32
25 October 2022	01:00	6.9	1539	0.01
25 October 2022	07:00	5.0	0	*∞*
25 October 2022	13:00	9.3	0.5	32
25 October 2022	19:00	6.4	2.67	6

## Data Availability

Data can be publicly accessed after logging at https://www.cyberforest.it/ (accessed on 13 March 2023).

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
