# Peer review of "Living Plants Ecosystem Sensing: A Quantum Bridge between Thermodynamics and Bioelectricity"

_biomimetics, 2023, doi:10.3390/biomimetics8010122_

Round 1
Reviewer 1 Report
-
Please refer to the attachment for modification suggestions in the article

Author Response
A. Chiolerio* G. Vitiello, M.M. Dehshibi, and A. Adamatzky,
We wish to thank the Editor for having coordinated the review of the manuscripts and the Reviewers for their precious time. We have collected all raised points in this response letter, providing punctual answers and references to changes made in the main text file. We took this opportunity to re-check the manuscript again and correct some grammatical inaccuracies. All changes are marked in red.
Reviewer 1
1. In this paper, the authors demonstrate the correlation between electrical signals and thermodynamics in spruce. It is suggested that bioelectrical signals can reflect the metabolic state of plants. Through experimental studies at different stages of plant development, it is concluded that the response of bioelectrical signals and temperature changes may be precisely synchronized, or may lag behind.
The results are a huge boost for plants as environmental sensors. At the
same time, the article is logical and standardized, but the paper still has some
shortcomings:
1. The summary of the abstract is not clear enough to express the main content and conclusions of the paper.
Response – Thank you for the time you have dedicated to our work and for
the useful suggestions. We have rephrased the abstract including your own
summary of the paper.
2. The pictures are not processed carefully enough. Delete the redundant external lines in pictures 2, 3, 4 and 6. Figure 2d lacks ordinate, and c and d are
changed to B-site in Figure 2. Figure 4b is changed to G-site, and S-G filtered
over 101 points. In Figure 4, Figures a and b are not aligned horizontally, and
the positions of c and d are reversed. The font in Figure 6 is too small, and the
legend font in 6a is a different size than the other fonts.
Response – Thanks for pointing this out. Figure 2, 3, 4 and 6 have been
reprocessed to delete the redundant lines. Figure 2d now shows ordinates, and
B-site inset is shown correctly. Figure 4b was changed to G-site, and S-G filtered
over 101 points. In Figure 4, Figures a and b were aligned horizontally, and the
positions of c and d was reversed. Figure 6 was organized in a different way,
now two panels stand in a row and the third, most significant panel, is centered
and magnified. The legend of panel 6a was adjusted.
3. Please add why there is a big difference in biopotential at different positions
of wood. Does Figure 2(b) correspond to the data at the same time period
in Figure 2( and why there is a big difference in the amplitude of biopotential
measured at the same position? Why are the minimum biological potentials at
- more uniform and similar than at other positions in Figure 2c?
Response – Thanks for pointing this out. About the difference in biopotentials
values, we should consider first of all that the scale is relative, i.e. we always
show the difference between two electrodes. Let us consider first the electrodes
vertically aligned (site B). For the couples α − β and ι − κ this difference is
about 100 to 120 mV, for the couples γ − δ and ϵ − ζ this difference is about
20 to 40 mV, most probably this small difference can also be influenced by
the contact resistance at the site of collection, which cannot be controlled. It
might be possible that the natural production of resin can increase contact
resistance and reduce the amplitude of the measured signals. Remarkably, the
couple η − θ shows the highest relative potential, around 650 mV, and this
could be due to the fact that water piezometric pressure in the soil creates an
ionic gradient whose boundary occurs at a certain height, here most probably
corresponding to the level of said electrodes. Dead logs (site E) show potential
differences of 70 to 80, 120 to 140, 20 to 30, 10 to 20 and 70 to 90 mV, therefore
with a less pronounced dispersion of values. For what concerns the electrodes
positioned radially around the trunk (site G), the higher circle shows potential
differences of about 80 to 100 mV, while the lower circle features potential
differences between 160 and 200 mV with the remarkable exception of the couple ι − κ, where the values are between 900 and 930 mV. The reason for this high discrepancy must be found in the position of the electrodes: those facing the sun, at the time of recordings, feature the highest potential difference, confirming that the physiological activity of the plant can be properly traced by sensors. Observations on amplitudes, frequencies and noise of the signal are obviously reflected in the complexity measures. This text has been also reported in the paper. The time scale of panels b in figures 2, 3 and 4 correspond to the
numerical analysis whose output is shown in panels c and d.
4. Figure 2, Figure 3 and Figure 4 are three groups of comparison tests, and
the units of independent variables should be consistent, such as 3c and 3d the
test time was changed from unit to min.
Response – Thanks for pointing this out. The numerical analysis is performed
using Matlab scripts on chunks of data whose size is 10 minutes. Therefore for
panels c and d we preferred to maintain the time in minutes, which is more
appropriate to the resolution of the curves. Biopotential measurements are on
2 the scale of seconds, as the sampling rate is 1 Hz. Now all three figures are
consistent.
5. Please add how the Kolmogorov complexity and fractal dimension data of
different positions of the three trees in the article are obtained. It is mentioned
in the article that saplings have more obvious bioelectrical activity, please give
specific data comparison explanation. Kolmogorov complexity is different at
different locations on the tree. Explain what it is related to, why it is different,
and why it is different at the same location.
Response – All of the complexity measures are calculated using the formulas
reported in Chapter 2.2 from the raw measurements, where each differential
channel generates one complexity curve, as shown in panels c and d of figures
2, 3 and 4. Saplings generate a spontaneous activity that produces jumps in
the biopotential recordings, this is particularly evident for example in Figure
4a, couple ϵ − ζ. Such jumps cannot be found in the more uniform noise of
the older tree (see Figure 2a) or the logs (Figure 3a). Kolmogorov entropy and
the other complexity measures are used to interpret in a more ”holistic” way
the biopotential curves. A higher entropy is due to a higher number of fluctuations in the biopotential per unit of time. Channels feature different number and amplitude of fluctuations in reason of the physiological activity that is generating each particular signal: electrical probes are point sensors that record
phenomena connected to a living being and its huge complexity. Differences
in entropy therefore are also generated by the photosynthetic activity, that is
more influenced by topmost electrodes, or by couples positioned closer to the
Sun position. This text was added to the paper.
6. The three groups of infrared thermal imaging images in Figure 5 are all tested
at the same time, so a row of time data can be reserved.
Response – Thanks for pointing this out. A row of time data was deleted and
the figure reorganized.
7. Figure 6 shows the correlation between thermodynamics and bioelectricity,
and the test process and data calculation process should be appropriately supplemented.
Figure 6a and Figure 6c lack error bars, and why are the values of
abscissa time different?
Response – The test process and data calculation is described now in Materials
and Methods section. We added error bars to panel 6c, while panel 6a blue curve does not have any error bar, as it represents the fractal dimension of only one electrode couple (η−θ). This is specified in the legend, wherever an average was computed and a standard deviation is available, the symbol < FD > appears.
The time in the abscissa is the real time when the thermography was acquired.
Consider that to move from one site to another, it takes some seconds to move
and position the thermal camera before the shot.
8. The article listed a large number of formulas, but the lack of relevant test
data and test process, theoretical analysis is not clear.
Response – Thanks for pointing this out. We have introduced in Section
4 a sentence where we have specified that our theoretical modelling aims to
the description and understanding of the results presented in Section 3 and
we also admit that the modelling provides at the moment only a qualitative
understanding, although it has the merit of framing the experimental results in
a unified theoretical view based on basic first principles. We have then listed the
main points of the results to be explained in terms of the theoretical mechanisms at the basis of our modelling. In each of the subsections, we have clarified how each of these points finds its understanding in the modelling. We have confined in the appendices most of the mathematical formalism to make the reading smooth and the presentation more transparent.
9. In the experiment of studying time correlation, compare the biopotential vector of site ”B” with that of sites ”E”, ”F” and ”G”, and what is the conclusion
reached? Please add clarification.
Response – Thanks for pointing this out. It is concluded that the response
of bioelectrical signals of different trees may be precisely synchronized, or may
lag behind, according to the particular hour of the day, or the metabolic status
of the forest, which in turn depends on the environmental conditions, such as
temperature, humidity, insolation, water presence. We added this text to the
paper.

Reviewer 2 Report
Minor revisions were suggested as below:
The font in Figure 6 is too small. Figure 6 can not be read well. Please revise it.
Author Response
A. Chiolerio* G. Vitiello, M.M. Dehshibi, and A. Adamatzky,
We wish to thank the Editor for having coordinated the review of the manuscripts and the Reviewers for their precious time. We have collected all raised points in this response letter, providing punctual answers and references to changes made in the main text file. We took this opportunity to re-check the manuscript again and correct some grammatical inaccuracies. All changes are marked in red.
Reviewer 2 The font in Figure 6 is too small. Figure 6 can not be read well.
Please revise it.
Response – Thanks for pointing this out. Figure 6 has been reconfigured and
should be more readable now.
